# Explaining the Behavior of Black-Box Prediction Algorithms with Causal Learning

**Numair Sani**[*]
*Sani Analytics, Mumbai, MH India*

*numairsani@gmail.com*

**Daniel Malinsky**[*]
*Department of Biostatistics, Columbia University, New York, NY USA*

*d.malinsky@columbia.edu*

**Ilya Shpitser**
*Department of Computer Science, Johns Hopkins University, Baltimore, MD USA*

*ilyas@cs.jhu.edu*

**Reviewed on OpenReview:** *https://openreview.net/forum?id=ZrqLpXbXvA*

## Abstract

Causal approaches to post-hoc explainability for black-box prediction models (e.g., deep neural networks trained on image pixel data) have become increasingly popular. However, existing approaches have two important shortcomings: (i) the "explanatory units" are micro-level inputs into the relevant prediction model, e.g., image pixels, rather than interpretable macro-level features that are more useful for understanding how to possibly change the algorithm's behavior, and (ii) existing approaches assume there exists no unmeasured confounding between features and target model predictions, which fails to hold when the explanatory units are macro-level variables. Our focus is on the important setting where the analyst has no access to the inner workings of the target prediction algorithm, rather only the ability to query the output of the model in response to a particular input. To provide causal explanations in such a setting, we propose to learn causal graphical representations that allow for arbitrary unmeasured confounding among features. We demonstrate the resulting graph can differentiate between interpretable features that causally influence model predictions versus those that are merely associated with model predictions due to confounding. Our approach is motivated by a counterfactual theory of causal explanation wherein good explanations point to factors that are "difference-makers" in an interventionist sense.

## 1 Introduction

In recent years, black-box artificial intelligence (AI) or machine learning (ML) prediction algorithms have exhibited impressive performance in a wide range of prediction tasks. In particular, methods based on deep neural networks (DNNs) have been successfully used to analyze high-dimensional data in settings such as healthcare and social data analytics (Esteva et al., 2019; Guimaraes et al., 2017). An important obstacle to widespread adoption of such methods, particularly in socially-impactful settings, is their black-box nature: it is not obvious, in many cases, how to explain the predictions produced by such algorithms when they succeed (or fail), given that they find imperceptible patterns among high-dimensional sets of features. Given the importance and consequences of these decisions, it is important to audit these algorithms to ensure they are fair, safe, and act in the best interests of the communities they affect.

To obtain impartial audits, it is desirable to have independent organizations or teams evaluate proposed prediction algorithms. This often means such audits must be performed without access to the inner workings of the target algorithm, due to intellectual property concerns. In such cases, post-hoc explainability methods

---

[*]Equal contribution.

that do not require access to the inner workings of the target algorithm are most appropriate. In this post-hoc setting, existing ML explainability approaches that involve retraining, modifying, or building newly explainable algorithms (Alvarez Melis & Jaakkola, 2018; Koh et al., 2020), or accessing intermediate layers (Kim et al., 2018; Bahadori & Heckerman, 2021; Yeh et al., 2020), are not directly applicable. "Feature importance" measures are popular; these aim to capture some generalized notion of association between each feature and a model's prediction output (or prediction error), but do not generally differentiate between features that are merely associated with the model prediction versus those that have a causal effect on the output. Distinguishing whether some variable is a cause of model outputs rather than only associated due to confounding is crucial to understanding whether manipulations of that feature will make any difference to the prediction model output; intervening on variables associated only due to confounding will have no consequence and provides no explanation of the outcome. We discuss this distinction and, more broadly, the importance of causal explainability in Section 2; see also Shin (2021) and Moraffah et al. (2020). Existing causality-aware feature importance approaches (Friedman, 2001; Loftus et al., 2024; Goldstein et al., 2015; Janzing et al., 2020) have important shortcomings in this post-hoc setting: (i) the explanatory features are micro-level inputs into the target prediction model, e.g., image pixels, rather than interpretable macro-level features that are more useful for understanding how to possibly change the algorithm's behavior, and (ii) existing approaches assume there exists no unmeasured confounding between features and target model predictions, which fails to hold when the explanatory units are macro-level variables. Throughout, we use "micro-level" to refer to the raw features directly input into the prediction model, which are typically pixel or voxel values in image analysis and may be words or tokens in text analysis. "Macro-level" features then correspond to (observed or latent) abstractions of the raw features that are independently manipulable, also described as causal representations or disentangled representations (Schölkopf et al., 2021).

Here we present an approach to post-hoc explanation in terms of user-selected macro-level interpretable features, building on ideas from the literature on causality and graphical models. By employing a causal graphical representation capable of dealing with unobserved confounding, we are able to provide valid causal explanations in terms of macro-level features and distinguish causal features from features merely associated with model predictions due to confounding. This is accomplished through the application of causal discovery methods (a.k.a. causal structure learning) (Spirtes et al., 2000; Pearl, 2009; Peters et al., 2017).

We begin by providing some background on causal explanation and causal modeling, the choice of "explanatory units", and the limitations of existing approaches. We then describe our causal discovery-based proposal for explaining the behaviors of black-box prediction algorithms. We present a simulation study that highlights key issues and demonstrates the strength of our approach. We apply a version of our proposal to two datasets: annotated image data for bird classification and annotated chest X-ray images for pneumonia detection. Finally, we discuss some applications, limitations, and future directions of this work.

## 2 Causal Explanation

There is a long history of debate in science and philosophy over what properly constitutes an explanation of some phenomenon. (In our case, the relevant phenomenon will be the output of a prediction algorithm.) A connection between explanation and "investigating causes" has been influential, in Western philosophy, at least since Aristotle. More recently, philosophical scholarship on *causal explanation* (Cartwright, 1979; Salmon, 1984; Woodward, 2005; Lombrozo & Vasilyeva, 2017) has highlighted various benefits to pursuing understanding of complex systems via causal or counterfactual knowledge, which may be of particular utility to the machine learning community. We focus here primarily on some relevant ideas discussed by Woodward (Woodward, 2005) to motivate our perspective in this paper, though similar issues are raised elsewhere in the literature.

In influential 20th-century philosophical accounts, explanation was construed via applications of deductive logical reasoning (i.e., showing how observations could be derived from physical laws and background conditions) or simple probabilistic reasoning (Hempel, 1965). One shortcoming – discussed by several philosophers in the late 20th-century – of all such proposals is that explanation is intuitively asymmetric: the height of a flagpole explains the length of its shadow (given the sun's position in the sky) but not vice versa; the length of a classic mechanical pendulum explains the device's period of motion, but not vice versa. Logical and

associational relationships do not exhibit such asymmetries. Moreover, some statements of fact or strong associations seem explanatorily irrelevant to a given phenomenon, as when the fact that somebody neglected to supply water to a rock "explains" why it is not living. (An analogous fact may have been more relevant for a plant, which in fact needs water to live.) Woodward argues that "explanatory relevance" is best understood via counterfactual contrasts and that the asymmetry of explanation reflects the role of causality.

On Woodward's counterfactual theory of causal explanation, explanations answer *what-would-have-been-different* questions. Specifically, the relevant counterfactuals describe the outcomes of interventions or manipulations. $X$ helps explain $Y$ if, under suitable background conditions, some intervention on $X$ produces a change in the distribution of $Y$. (Here we presume the object of explanation to be the random variable $Y$, not a specific value or event $Y = y$. That is, we choose to focus on *type-level* explanation rather than *token-level* explanations of particular events. See Beckers (2022) for related discussion focused on causal explanation at the event level.) This perspective has strong connections to the literature on causal models in artificial intelligence and statistics (Spirtes et al., 2000; Pearl, 2009; Peters et al., 2017). A causal model for outcome $Y$ precisely stipulates how $Y$ would change under various interventions. So, to explain black-box algorithms we endeavor to build causal models for their behaviors. We propose that such causal explanations can be useful for algorithm evaluation and informing decision-making. In contrast, purely associational measures will be symmetric, include potentially irrelevant information, and fail to support (interventionist) counterfactual reasoning.

Despite a paucity of causal approaches to explainability in the ML literature (with some exceptions, discussed later), survey research suggests that causal explanations are of particular interest to industry practitioners; Bhatt et al. (2020) quote one chief scientist as saying "Figuring out causal factors is the holy grail of explainability," and report similar sentiments expressed by many organizations.

## 3 Causal Modeling

Here we provide some background on causal modeling to make our proposal more precise. Throughout, we use uppercase letters (e.g., $X, Y$) to denote random variables or vectors and lowercase $(x, y)$ to denote fixed values.

We use causal graphs to represent causal relations among random variables (Spirtes et al., 2000; Pearl, 2009). In a causal directed acyclic graph (DAG) $\mathcal{G} = (V, E)$, a directed edge between variables $X \rightarrow Y$ $(X, Y \in V)$ denotes that $X$ is a direct cause of $Y$, relative to the variables on the graph. Direct causation may be explicated via a system of nonparametric structural equations (NPSEMs) a.k.a. a structural causal model (SCM). The distribution of $Y$ given an intervention that sets $X$ to $x$ is denoted $p(y \mid \mathrm{do}(x))$ by Pearl (Pearl, 2009). Causal effects are often defined as interventional contrasts, e.g., the average causal effect (ACE): $\mathbb{E}[Y \mid \mathrm{do}(x)] - \mathbb{E}[Y \mid \mathrm{do}(x')]$ for values $x, x'$. Equivalently, one may express causal effects within the formalism of potential outcomes or counterfactual random variables, c.f. (Richardson & Robins, 2013).

Given some collection of variables $V$ and observational data on $V$, one may endeavor to learn the causal structure, i.e., to select a causal graph supported by the data. We focus on learning causal relations from purely observational (non-experimental) data here, though in some ML settings there exists the capacity to "simulate" interventions directly, which may be even more informative. There exists a significant literature on selecting causal graphs from a mix of observational and interventional data, e.g. Triantafillou & Tsamardinos (2015); Wang et al. (2017b), and though we do not make use of such methods here, the approach we propose could be applied in those mixed settings as well.

There are a variety of algorithms for causal structure learning, but what most approaches share is that they exploit patterns of statistical constraints implied by distinct causal models to distinguish among candidate graphs. One paradigm is constraint-based learning, which will be our focus. In constraint-based learning, the aim is to select a causal graph or set of causal graphs consistent with observed data by directly testing a sequence of conditional independence hypotheses. Distinct models will imply distinct patterns of conditional independence, and so by rejecting (or failing to reject) a collection of independence hypotheses, one may narrow down the set of models consistent with the data. (These methods typically rely on some version of the faithfulness assumption (Spirtes et al., 2000), which stipulates that all observed independence constraints

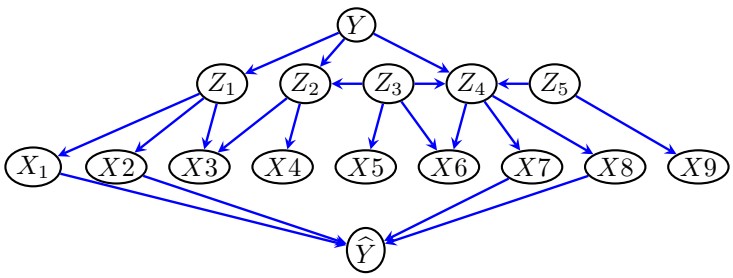

Figure 1: An example DAG to represent an image data-generating process. $Y$ denotes the true label (e.g., disease status), $Z$s denote high-level interpretable features, and $X$s denote (clusters or collections of) pixels. The output of the prediction algorithm trained on $X$ is $\widehat{Y} = f(X)$.

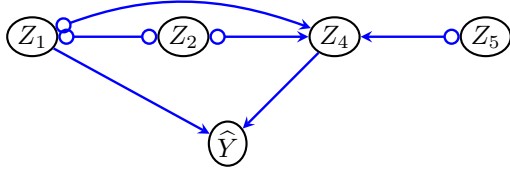

Figure 2: The PAG representation of the process above, with vertices $(Z_1, Z_2, Z_4, Z_5, \widehat{Y})$.

correspond to missing edges in the graph.) For example, a classic constraint-based method is the PC algorithm (Spirtes et al., 2000), which aims to learn an equivalence class of DAGs by starting from a fully-connected model and removing edges when conditional independence constraints are discovered via statistical tests. Since multiple DAGs may imply the same set of conditional independence constraints, PC estimates a CPDAG (completed partial DAG), a mixed graph with directed and undirected edges that represents a Markov equivalence class of DAGs. (Two graphs are called Markov equivalent if they imply the same conditional independence constraints.) Variations on the PC algorithm and related approaches to selecting CPDAGs have been thoroughly studied in the literature (Colombo & Maathuis, 2014; Cui et al., 2016).

In settings with unmeasured (latent) confounding variables, it is typical to study graphs with bidirected edges to represent dependence due to confounding. For example, a partial ancestral graph (PAG) (Zhang, 2008a) is a graphical representation which includes directed edges ($X \rightarrow Y$ means $X$ is a causal ancestor of $Y$), bidirected edges ($X \leftrightarrow Y$ means $X$ and $Y$ are both caused by some unmeasured common factor(s), e.g., $X \leftarrow U \rightarrow Y$), and partially directed edges ($X \circ\!\!\rightarrow Y$ or $X \circ\!\!-\!\!\circ Y$) where the circle marks indicate ambiguity about whether the endpoints are arrowheads or tails. Generally, PAGs may also include additional edge types to represent selection bias, but this is irrelevant for our purposes here. PAGs inherit their causal interpretation by encoding the commonalities among a set of underlying causal DAGs with latent variables. A bit more formally, a PAG represents an equivalence class of maximal ancestral graphs (MAGs), which encode the independence relations among observed variables when some variables are unobserved (Richardson & Spirtes, 2002; Zhang, 2008a). The FCI algorithm (Spirtes et al., 2000; Zhang, 2008b) is a well-known constraint-based method, which uses sequential tests of conditional independence to select a PAG from data. Similarly to the PC algorithm, FCI begins by iteratively deleting edges from a fully-connected graph by a sequence of conditional independence tests. However, the space of possible models (mixed graphs with multiple types of edges) is much greater for FCI as compared with PC, so the rules linking patterns of association to possible causal structures are more complex. Though only independence relations among observed variables are testable, it can be shown formally that certain patterns of conditional independence among observed variables are incompatible with certain latent structures (assuming faithfulness), so some observed patterns will rule out unmeasured confounding and other observed patterns will on the contrary suggest the existence of unmeasured confounders. Details of the FCI algorithm may be found in the literature (Zhang, 2008b; Colombo et al., 2012). Variations on the FCI algorithm and alternative PAG-learning procedures have also been studied (Colombo et al., 2012; Claassen & Heskes, 2012; Ogarrio et al., 2016; Tsirlis et al., 2018).

### 3.1   Explanatory Units

Consider a supervised learning setting with a high-dimensional set of "low-level" features (e.g., pixels in an image) $X = (X_1, ..., X_q)$ taking values in an input space $\mathcal{X}^q$ and outcome $Y$ taking values in $\mathcal{Y}$. A prediction or classification algorithm (e.g., DNN) learns a map $f : \mathcal{X}^q \mapsto \mathcal{Y}$. Predicted values from this function are denoted $\widehat{Y}$. To explain the predictions $\widehat{Y}$, we focus on a smaller set of "high-level" features $Z = (Z_1, ..., Z_p)$ $(p \ll q)$ that are "interpretable" in the sense that they correspond to human-understandable and potentially manipulable elements of the application domain of substantive interest, e.g., "effusion in the lung," "presence of tumor in the upper lobe," and so on in lung imaging or "has red colored wing feathers," "has a curved beak," and so on in bird classification. Though we use mostly examples from image classification in our discussion and hence the language of "pixels," the raw input data may instead consist in text (words or tokens), measurements taken over time, or other kinds of "low-level" signals.

Though the target prediction algorithm takes low-level features $X = (X_1, ..., X_q)$ as input, our interest is explaining the output $\widehat{Y}$ in terms of $Z = (Z_1, ..., Z_p)$. One reason is that individual micro-features (e.g. pixels) may make very little causal difference to the output of a prediction model, but have important effects in aggregate. That is, groups of pixels (not necessarily spatially contiguous) or higher-level statistical properties of pixels (e.g., the variance in brightness of some region of pixel space, the existence of shapes, borders, or other contrasts) are often truly what make a difference to a prediction algorithm. A paradigmatic example of this is when the background color or lighting of a photograph has a strong effect on the predicted label – intervening to change an individual background pixel has no consequence, but setting the photographic subject against a different background or in different lighting conditions may change the output dramatically. This highlights a second reason to focus on macro-level interpretable features: they more often coincide with relevant manipulable elements of the research domain.

### 3.2   Induced Unmeasured Confounding

When the relevant explanatory units do not coincide with the set of raw features used by the prediction algorithm, unmeasured confounding becomes a salient issue. In particular, if the set $Z = (Z_1, ..., Z_p)$ is selected by a user it is not generally possible to know if "all the relevant features" are included in $Z$. There is always the possibility that some causally-important macro-level features have been excluded (as would be the case if the user did not know a priori that "lighting conditions" had an important effect on the prediction output). The data-generating process we assume may be represented by a graph like the one in Figure 1. Here $Y$ denotes the true concept or label that we aim to predict (e.g., disease status of a patient). This generates the high-level features $Z$ (e.g., symptoms or findings on an X-ray), which are rendered via imaging/data recording technology as pixels $X$. The output of the prediction algorithm $\widehat{Y}$ is a direct function of only raw inputs $X$. Importantly, the elements of $Z$ may be causally related to each other: e.g., in a medical setting, interventions on some symptoms may lead to changes in other symptoms. Thus there may be directed edges from some $Z_i \to Z_j$ as in Figure 1. Note that in the hypothetical model depicted here, some elements of $Z$ have causal pathways to $\widehat{Y}$ and others do not, and some elements may appear associated with $\widehat{Y}$ despite no causal pathway. ($Z_1$ has a causal pathway to $\widehat{Y}$, while $Z_2$ does not. Yet, $Z_1$ and $Z_2$ are associated due to their common parent $Y$, so $\widehat{Y}$ and $Z_2$ will likely be associated in the data.) In applications, the true underlying data-generating DAG is unknown: we may have substantial uncertainty about both how the macro-level variables are related to each other, which pixels or groups of pixels they affect, and which pixels play an important role in the predicted output $\widehat{Y} = f(X)$. Causal discovery algorithms may thus be illuminating here, in particular causal discovery algorithms that are consistent in the presence of unmeasured confounding.

### 3.3   Limitations Of Existing Methods

Given this emphasis on macro-level explanatory variables and the attendant issue of unmeasured confounding, existing approaches to explainability have important limitations. One popular class of methods are "perturbation-based" approaches (Chattopadhyay et al., 2019; Schwab & Karlen, 2019; Selvaraju et al., 2017; Simonyan et al., 2013; Shrikumar et al., 2017; Chen et al., 2024), which perturb each element $X_i$,

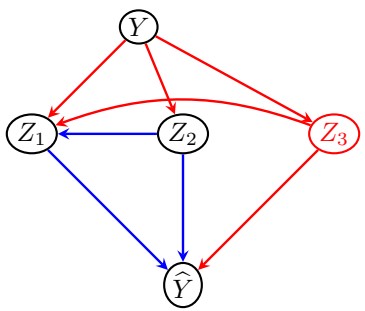

Figure 3: Data generating process for Example 1.

holding the remaining elements $X_{-i}$ fixed and observe the change in $\widehat{Y}$. Since the relationships among the $X$s and $Z$s are complex and unknown, these perturbation-based quantities cannot be straightforwardly translated into explanations as a function of macro-level explanatory units ($Z$s). Some causality-aware feature importance approaches, such as Partial Dependence Plots (Friedman, 2001), Causal Dependance Plots (Loftus et al., 2024), Individual Conditional Expectation plots (Goldstein et al., 2015), and modified causal variations of classical Shapley values (Janzing et al., 2020; Jung et al., 2022; Heskes et al., 2020) may be applied (or adapted) to provide explanatory information at the level of macro variables. However, these approaches assume there is no unmeasured confounding for the chosen set of explanatory features. Ultimately, these approaches will estimate quantities that adjust for (condition on) some subset of the observed features, but in the presence of unmeasured confounding there may be no observed subset that is valid. This can lead to substantially incorrect "explanatory" conclusions.

We illustrate this latter point with a simple example using Shapley values (Lundberg & Lee, 2017). Shapley values are a feature attribution approach with roots in game theory. Given a game with $N$ players and a cost function $v(N)$, Shapley values allocate cost to each player by defining a value $\Phi_i$ for each player $i$, with $\Phi_i$ computed as

$$\Phi_i = \sum_{S \subseteq N \setminus \{i\}} \frac{|S|!(|N| - |S| - 1)}{|N|!} [v(S \cup \{i\}) - v(S)]$$

The Shapley values satisfy the property $v(N) = \sum_{i=1}^{N} \Phi_i$. Important criticisms of Shapley values as feature importance measures have been raised elsewhere (Kumar et al., 2020). We examine a modified approach proposed in Janzing et al. (2020) that relies on backdoor adjustment (Pearl, 2009) to perform causal attribution of the prediction made by a model to its individual features. This amounts to a causally-aware proposal for estimating the contributions $v(S \cup \{i\})$ in the setting where "costs" are model predictions and "players" are features. Though the authors in Janzing et al. (2020) mainly consider a setting wherein the inputs to the target prediction algorithm correspond to the explanatory units, here we consider a version of the approach applied to macro-level features $Z = (Z_1, ..., Z_p)$. (If there is a mismatch between model inputs and explanatory units, it is not straightforward to apply the approach described in Janzing et al. (2020).) In the presence of unmeasured confounding, it is possible that the Shapley value calculated following Janzing et al. (2020) for a particular feature is zero even when the feature is in fact causally relevant to the prediction model output. We demonstrate this through a brief example here, with the relevant calculations presented in the Appendix.

**Example 1**: Consider the causal graph shown in Figure 3, associated with the following SCM:

$$Y = \epsilon_Y$$
$$Z_3 = \gamma_0 + \gamma_1 Y + \epsilon_{Z_3}$$
$$Z_2 = \theta_0 + \theta_1 Y + \epsilon_{Z_2}$$
$$Z_1 = \nu_0 + \nu_1 Z_2 + \nu_2 Z_3 + \nu_3 Y + \epsilon_{Z_1}$$
$$\widehat{Y} = \mu_0 Z_1 Z_3 + \mu_1 Z_1 + \mu_2 Z_2$$

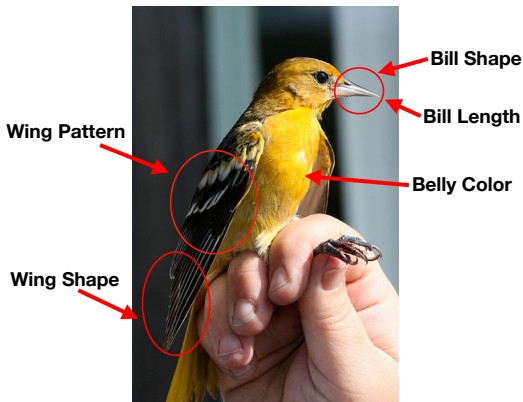

Figure 4: An image of a Baltimore Oriole annotated with interpretable features.

The Shapley value for $Z_1$ calculated using a valid adjustment set $(Z_2, Z_3)$ may be calculated as:

$$\phi_{Z_1} = \frac{0!2!}{3!}\left\{f'_{\{Z_1\}} - f'_\phi\right\} + \frac{1!1!}{3!}\left\{f'_{Z_1,Z_2} - f'_{Z_2}\right\}$$
$$+ \frac{1!1!}{3!}\left\{f'_{Z_1,Z_3} - f'_{Z_3}\right\} + \frac{2!0!}{3!}\left\{f'_{\{Z_1,Z_2,Z_3\}} - f'_{\{Z_2,Z_3\}}\right\}.$$

This leads to the following Shapley value for $Z_1$:

$$\phi_{Z_1} = \frac{1}{2}\left\{\mu_0 z_1 \mathbb{E}[Z_3] - \mu_0\mathbb{E}[Z_1 Z_3] + \mu_0 z_1 z_3 - \mu_0 z_3 \mathbb{E}[Z_1]\right\}$$
$$+ \mu_1\left\{z_1 - \mathbb{E}[Z_1]\right\}.$$

However, if $Z_3$ is omitted from the chosen set of explanatory features, backdoor adjustment can only be (incorrectly) performed using $Z_2$. In that case, the Shapley value for $Z_1$ would be:

$$\phi_{Z_1} = (z_1 - \mathbb{E}[Z_1])[\mu_0 z_3 + \mu_1]$$

Detailed calculations may be found in the Appendix. Now, if $z_1 = \mathbb{E}[Z_1]$, then the incorrect Shapley value will be 0, while the true Shapley value will be $\frac{1}{2}\left\{\mu_0\mathbb{E}[Z_1]\mathbb{E}[Z_3] - \mu_0\mathbb{E}[Z_1 Z_3]\right\}$, which can be non-zero. This illustrates a key shortcoming of Shapley values in this setting: if there is unmeasured confounding, even causally-informed Shapley values may lead to misleading conclusions about feature importance. There are promising recent proposals combining Shapley with identification theory for mixed graphs representing confounded variable sets (Jung et al., 2022), but in the setting considered here, the correct graph is not known.

## 4    Explaining Black-Box Predictions

Recall that our task is to explain the behavior of some black-box prediction model trained on features $X = (X_1, ..., X_q)$ by identifying the causal determinants of $\widehat{Y} = f(X)$ from among a set of macro-level features $Z = (Z_1, ..., Z_p)$.

For now, we assume that the interpretable feature set is predetermined and available in the form of an annotated dataset, i.e., that each image is labeled with interpretable features. Later, we discuss the possibility of automatically extracting a set of possibly interpretable features from data which lacks annotations. We

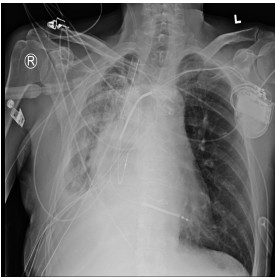

Figure 5: A chest X-ray from a pneumonia patient. The image was annotated by a radiologist to indicate `Effusion`, `Pneumothorax`, and `Pneumonia`.

allow that elements of $Z$ are statistically and causally interrelated, but assume that $Z$ contains no variables which are deterministically related (e.g., where $Z_i = z_i$ always implies $Z_j = z_j$ for some $i \neq j$), though we also return to this issue later. Examples from two image datasets that we discuss later, on bird classification and lung imaging, are displayed in Figures 4 and 5: $X$ consists of the raw pixels and $Z$ includes the interpretable annotations identified in the figures.

Here we describe the assumed data-generating process more formally. We use $\mathrm{Pa}(V, \mathcal{G})$ to denote the set of parents of a vertex $V$ in graph $\mathcal{G}$, i.e., vertices with directed edges into $V$. Let $\mathcal{G}$ be a DAG with vertices $Y, Z^*, X, \widehat{Y}$ such that $\mathrm{Pa}(Y, \mathcal{G}) = \emptyset$ (the "true label" has no causal parents), $\mathrm{Pa}(Z_i^*, \mathcal{G}) \subseteq \{Z_{-i}^*, Y\}\ \forall i$ (each macro-level feature may be caused by other macro-level features as well as $Y$), $\mathrm{Pa}(X_j, \mathcal{G}) \subseteq Z^*\ \forall j$ (micro-level features are determined by macro-level features), and $\mathrm{Pa}(\widehat{Y}, \mathcal{G}) \subseteq X$ (the model predictions are a function of model inputs). Let $Z \subseteq Z^*$ denote the subset of macro-level features that are recorded in the data. Described as a SCM, we assume the data-generating process has the following form:

$$
\begin{aligned}
Y &\sim P \\
Z_i^* &= g_i(\mathrm{Pa}(Z_i^*, \mathcal{G}), \epsilon_i) \quad \forall i \\
X_j &= h_j(\mathrm{Pa}(X_j, \mathcal{G}), \nu_j) \quad \forall j \\
\widehat{Y} &= f(\mathrm{Pa}(\widehat{Y}, \mathcal{G}))
\end{aligned}
$$

with all errors $\epsilon_i, \nu_j$ mutually independent. The marginal distribution over variables $(Z, \widehat{Y})$ satisfies the Markov condition wrt a MAG $\mathcal{M}$ (a projection of the DAG $\mathcal{G}$ onto this subset of vertices). The Markov equivalence class of $\mathcal{M}$ is represented by PAG $\mathcal{P}$ (Zhang, 2008a).

Our proposal is to estimate a causal PAG $\widehat{\mathcal{G}}$ over the variable set $V = (Z, \widehat{Y})$, with the minimal background knowledge imposed that the predicted outcome $\widehat{Y}$ is a causal non-ancestor of $Z$ (there is no directed path from $\widehat{Y}$ into any element of $Z$). This knowledge simply reflects the fact that the output of a prediction algorithm trained on an image does not cause the content of that image.

We primarily use the FCI algorithm in our experiments, but alternative PAG-learning discovery algorithms may be also be used. FCI selects a PAG by executing the following steps: (1) beginning with a complete graph over the vertices, a series of conditional independence hypotheses are tested for every pair of verticies and every possible conditioning set. When an independence is discovered between two vertices, the edge between them is removed and the relevant conditioning set is recorded (the separating set). (2) The procedure then orients some triples of variables as colliders using the so-called "collider rule": a triple $Z_i \circ\!\!-\!\!\circ Z_j \circ\!\!-\!\!\circ Z_k$ with $Z_i$ and $Z_k$ not adjacent is oriented $Z_i \circ\!\!\rightarrow Z_j \leftarrow\!\!\circ Z_k$ only if $Z_j$ is not in the separating set for $Z_i, Z_k$. The result at this stage is a partially-oriented graph. (3) Some additional independence tests are carried out based on the graph structure (see detailed discussion in Colombo et al. (2012)) and the collider rule is repeated. (4) Finally, some additional orientations may follow from the orientations already determined, using provably complete graphical rules that are applied exhuastively to the partially oriented graph (Zhang, 2008b). Assuming faithfulness, the FCI algorithm is an asymptotically correct estimator of $\mathcal{P}$ (Spirtes et al., 2000; Zhang, 2008b; Colombo et al., 2012). Simulations previously reported indicate that when the ground

Table 1: Types of PAG edges relating some interpretable feature $Z_i$ and the predicted outcome $\widehat{Y}$.

| EDGE TYPE | EDGE INTERPRETATION |
|---|---|
| $Z_i \to \widehat{Y}$ | $Z_i$ is a cause of $\widehat{Y}$ |
| $Z_i \leftrightarrow \widehat{Y}$ | $Z_i$ and $\widehat{Y}$ share an unmeasured common cause $Z_i \leftarrow U \to \widehat{Y}$ |
| $Z_i \circ\!\!\to \widehat{Y}$ | Either $Z_i$ is a cause of $\widehat{Y}$ or there is unmeasured confounding, or both |

truth is sparse and the sample size is sufficiently large, FCI performs well and recovers a structure close to the true PAG. However, every conditional independence test performed has some chance of error, and compounding errors may lead to the incorrect structure especially when sample size is small or the truth is dense. Though it is very difficult to exactly recover the correct graph with any finite sample size, simulation studies show that graphs with good precision can in many cases be estimated from on the order of a few thousand samples (Colombo et al., 2012; Ogarrio et al., 2016).

Additional background knowledge may also be imposed if it is known, for example, that none of the elements of $Z$ may cause each other (they are only dependent due to latent common causes) or there are groups of variables which precede others in a causal ordering. If in the estimated graph $\widehat{\mathcal{G}}$, there is a directed edge $Z_i \to \widehat{Y}$, then $Z_i$ is a cause (definite causal ancestor) of the prediction, if instead there is a bidirected edge $Z_i \leftrightarrow \widehat{Y}$ then $Z_i$ is *not* a cause of the prediction but they are dependent due to common latent factors, and if $Z_i \circ\!\!\to \widehat{Y}$ then $Z_i$ is a possible cause (possible ancestor) of the prediction but unmeasured confounding cannot be ruled out. These edge types are summarized in Table 1. The reason it is important to search for a PAG and not a DAG (or equivalence class of DAGs) is that $Z$ will in general *not* include all possibly relevant variables.

## 5   A Simulation Study

To illustrate our proposed approach and its ability to distinguish truly causal features from "spurious" (confounded) features, we carry out the following experiment, inspired by and modified from a study reported in Chalupka et al. (2015). Black-and-white images are generated containing various geometric shapes (alone or in combination): a horizontal bar ($H$), vertical bar ($V$), circle ($C$), or triangle ($T$), in addition to random pixel noise; see Figure 6. The true binary label $Y$ is a Bernoulli random variable which makes the appearance of certain shapes more or less likely. Data is generated according to the DAG in Figure 7(a), with the following parameters used in the data generating process:

$$Y \sim \text{Bernoulli}(0.4)$$
$$C \sim \text{Bernoulli}(0.25)$$
$$V \sim \text{Bernoulli}(\text{expit}(-1.13 + 2.3 \times Y))$$
$$H \sim \text{Bernoulli}(\text{expit}(-2.64 + 3.112 \times V + 1.5974 \times Y))$$
$$T \sim \text{Bernoulli}(\text{expit}(-2 + 1.3 \times C + 2.2 \times H))$$

The task is to predict the binary label $Y$ on the basis of raw image pixel data. We build a black-box model based on a multiclass neural network architecture called HydraNet (Cohen, 2022). In order to control, for the purposes of this illustrative simulation, which variables are truly causes of the prediction output, we combine this neural network with logistic regression. We first pre-train HydraNet on a dataset of 2,500 images to predict the presence of horizontal bars, vertical bars, triangles, and circles in a given image. Then, we train a logistic regression classifier using the predictions from HydraNet as the features and the label $Y$ as target, on a dataset of 20,000 images simulated according to the above DGP. An inspection of the logistic regression coefficients confirms that $\widehat{Y}$ is a function of only $V$ and $H$ – these are the true causes of the classifier's output, as illustrated graphically in Figure 7(a). We emphasize that the function of the final logistic regression in this prediction pipeline is only to constrain the predictions $\widehat{Y}$ to depend entirely on $V$ and $H$. Relevant details and calculations may be found in Appendix.

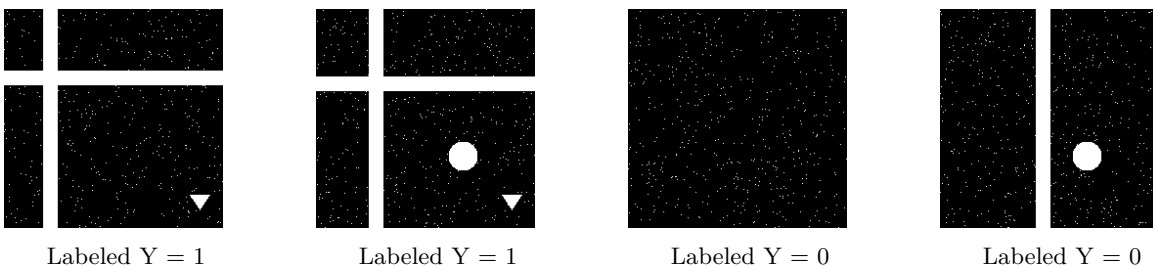

| Labeled Y = 1 | Labeled Y = 1 | Labeled Y = 0 | Labeled Y = 0 |

Figure 6: Simulated image examples with horizontal bars, vertical bars, circles, and triangles.

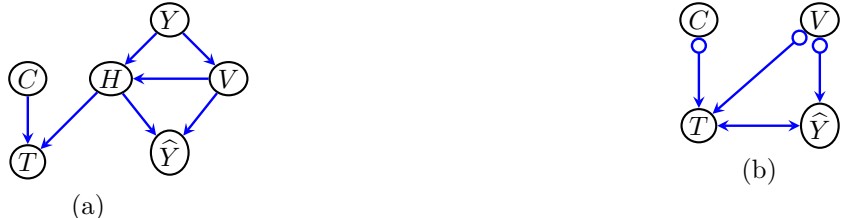

Figure 7: (a) A causal diagram representing the true data generating process. Here $Y$ denotes the true binary label and $C, T, H,$ and $V$ indicate the presence of circles, triangles, horizontal bars, and vertical bars in the images. $\widehat{Y}$ denotes the output of the combined neural network and logistic regression classifier. (b) The PAG learned using FCI over $(C, T, V, \widehat{Y})$.

The model performs reasonably well: 77.89% accuracy on a set of 20,000 images. Our interpretable features $Z$ are indicators for the presence of the various shapes in the image. Since the true underlying behavior is causally determined by $V$ and $H$, we expect $V$ and $H$ to be "important" for the predictions $\widehat{Y}$, but due to the mutual dependence the other shapes are also highly correlated with $\widehat{Y}$. Moreover, we mimic a setting where $H$ is (wrongfully) omitted from the set of candidate interpretable features; in practice, the feature set proposed by domain experts or available in the annotated data will typically exclude various relevant determinants of the underlying outcome. In that case, $H$ is an unmeasured confounder. Applying the PAG-estimation algorithm FCI to variable set $(C, T, V, \widehat{Y})$ we learn the structure in Figure 7(b), which indicates correctly that $V$ is a (possible) cause, but that $T$ is not: $T$ is associated with the prediction outcomes only due to an unmeasured common cause. (FCI incorporates the user-specified background knowledge that $\widehat{Y}$ is not a cause of any image features.)[1] This simple example illustrates how the estimated PAG using incomplete interpretable features can be useful. We disentangle mere statistical associations from (potentially) causal relationships, in this case indicating correctly that interventions on $V$ may make a difference to the prediction algorithm's behavior but interventions on $T$ would not, and moreover that there is a potentially important cause of the output ($H$) that has been excluded from our set of candidate features, as evident from the bidirected edge $T \leftrightarrow \widehat{Y}$ learned by FCI. Existing approaches that do not properly account for unmeasured confounding cannot distinguish between such features and would incorrectly identify $T$ as "important" for explaining $\widehat{Y}$.

## 6   Experiments: Bird Classification and Pneumonia Detection from X-rays

We conduct two real data experiments to demonstrate the utility of our approach. First, we study a neural network for bird classification, trained on the Caltech-UCSD 200-2011 image dataset (Wah et al., 2011). It consists of 200 categories of birds, with 11,788 images in total. Each image comes annotated with 312 binary attributes describing interpretable bird characteristics like eye color, size, wing shape, etc. We build a black-box prediction model based on a convolutional neural network (CNN) using raw pixel features to predict

---

[1]Throughout, we use FCI as implemented in the command-line interface to the TETRAD freeware: https://github.com/cmu-phil/tetrad.

the class of the bird and then use FCI to explain the output of the model. Second, we follow essentially the same procedure to explain the behavior of a pneumonia detection neural network, trained on a subset of the ChestX-ray8 dataset (Wang et al., 2017a). The dataset consists of 108,948 frontal X-ray images of 32,217 patients, along with corresponding free-text X-ray reports generated by certified radiologists. Both data sources are publicly available online.

## 6.1 Data Preprocessing & Model Training

### Bird Image Data

Since many of the species classifications have few associated images, we first broadly group species into 9 coarser groups. For example, we group the Baird Sparrow and Black Throated Sparrow into one Sparrow class. Details on the grouping scheme can be found in the Appendix. This leads to 9 possible outcome labels and 5514 images (instances that do not fit into one of these categories are excluded from analysis). The number of images across each class is not evenly balanced. So, we subsample overrepresented classes in order to get roughly equal number of images per class. This yields a dataset of 3538 images, which is then partitioned into training, validation, and testing datasets of 2489, 520, and 529 images respectively. We train the ResNet18 architecture (pre-trained on ImageNet) (He et al., 2016) on our dataset and achieve an accuracy of 86.57% on the testing set. For computational convenience and statistical efficiency, we consolidate the 312 available binary attributes into ordinal attributes. (With too many attributes, some combinations of values may rarely or never co-occur, which leads to data fragmentation.) For example, four binary attributes describing "back pattern" (solid, spotted, striped, or multi-colored), are consolidated into one attribute `Back Pattern` taking on five possible values: the 4 designations above and an additional category if the attribute is missing. Even once this consolidation is performed, there still exist many attributes with a large number of possible values, so we group together similar values. For example, we group dark colors such as gray, black, and brown into one category and warm colors such as red, yellow, and orange into another. Other attributes are consolidated similarly, as described in the Appendix. After the above preprocessing, for each image we have a predicted label from the CNN and 26 ordinal attributes.

### Chest X-ray Data

From the full ChestX-ray8 dataset, we create a smaller dataset of chest X-rays labeled with a binary diagnosis outcome: pneumonia or not. The original dataset contained 120 images annotated by a board-certified radiologist to contain pneumonia. To obtain "control" images, we randomly select 120 images with "no findings" reported in the radiology report. To generate interpretable features, we choose the following 7 text-mined radiologist finding labels: `Cardiomegaly`, `Atelectasis`, `Effusion`, `Infiltration`, `Mass`, `Nodule`, and `Pneumothorax`. If the radiology report contained one of these findings, the corresponding label was given a value 1 and 0 otherwise. This produces an analysis dataset of 240 images: 120 pneumonia and 120 "control" with each image containing 7 binary interpretable features. Using the same architecture as for the previous experiment and reserving 55 images for testing, ResNet18 achieves an accuracy of 74.55%.

## 6.2 Structure Learning

Structure learning methods such as FCI produce a single estimated PAG as output. In applications, it is common to repeat the graph estimation on multiple bootstraps or subsamples of the original data, in order to control false discovery rates or to mitigate the cumulative effect of statistical errors in the independence tests (Stekhoven et al., 2012). We create 50 bootstrapped replicates of the bird image dataset and 100 replicates of the X-ray dataset. We run FCI on each replicate with independence test rejection threshold (a tuning parameter) set to $\alpha = .05$ and $\alpha = .01$ for the birds and X-ray datasets, respectively, with the knowledge constraint imposed that outcome $\widehat{Y}$ cannot cause any of the interpretable features. Here FCI is used with the $\chi^2$ independence test, and we limit the maximum conditioning set size to 4 for computational tractability in the birds dataset. (No limit is necessary in the smaller X-ray dataset.) As a sensitivity analysis, we repeat both experiments using a different PAG-learning procedure, a hybrid algorithm that combines the greedy score-based procedure GRaSP (Lam et al., 2022) with FCI. The details are described in Appendix C.

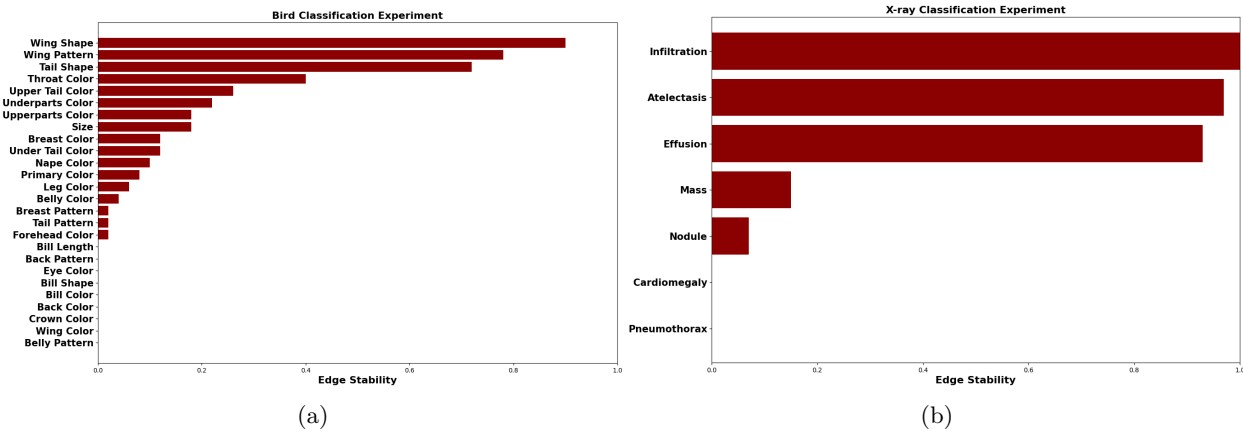

Figure 8: Results for two experiments. Potential causal determinants of (a) bird classifier output from bird images and (b) pneumonia classifier output from X-ray images.

## 6.3 Results

We compute the relative frequency over bootstrap replicates of $Z_i \to \widehat{Y}$ and $Z_i \circ\!\!\to \widehat{Y}$ edges from all attributes. This represents the frequency with which an attribute is determined to be a cause or possible cause of the predicted outcome label and constitutes a rough measure of "edge stability" or confidence in that attribute's causal status. The computed edge stabilities are presented in Figure 8. In the bird classification experiment, we find that the most stable (possible) causes include `Wing Shape`, `Wing Pattern`, and `Tail Shape`, which are intuitively salient features for distinguishing among bird categories. We have lower confidence that the other features are possible causes of $\widehat{Y}$. In the X-ray experiment, we find that edges from `Atelectasis`, `Infiltration`, and `Effusion` are stable possible causes of the `Pneumonia` label. This is reassuring, since these are clinically relevant conditions for patients with pneumonia. An estimated PAG from one of the bootstrapped subsamples is displayed in Figure 10. The FCI procedure has detected a variety of edge types, including bidirected ($\leftrightarrow$) edges to indicate unmeasured confounding among some variables, but edges from `Atelectasis`, `Infiltration`, and `Effusion` are consistently either directed ($\to$) or partially directed ($\circ\!\!\to$) into the predicted outcome.

Our sensitivity analysis using an alternative PAG-learning procedure, GRaSP-FCI, produces similar results. In the X-ray experiment, the final ordering of features by edge stability GRaSP-FCI is the same as the ordering produced by FCI. In the bird classification experiment, there are some substantive differences: a few features ranked highly by FCI are ranked low by GRaSP-FCI and vice versa, though for most features the conclusions are concordant. `Crown Color` and `Size` are ranked highest. The full results are presented in Appendix C.

## 6.4 Comparison to Pixel-Level Explainability Approaches

Having discussed some shortcomings of existing approaches in Section 3.3, we apply two popular pixel-level explanatory approaches to the data from each experiment. We first examine Locally Interpretable Model Explanations (LIME), proposed in Ribeiro et al. (2016). LIME uses an interpretable representation class provided by the user (e.g., linear models or decision trees) to train an interpretable model that is locally faithful to any black-box classifier. The output of LIME in our image classification setting is, for each input image, a map of super-pixels highlighted according to their positive or negative contribution toward the class prediction. The second approach we investigate is a version of the Shapley Additive Explanation (SHAP) algorithm, proposed by Lundberg & Lee (2017). The output of this method in our setting is a list of SHAP values for input features in the image that can be used to understand how the features contribute to a prediction.

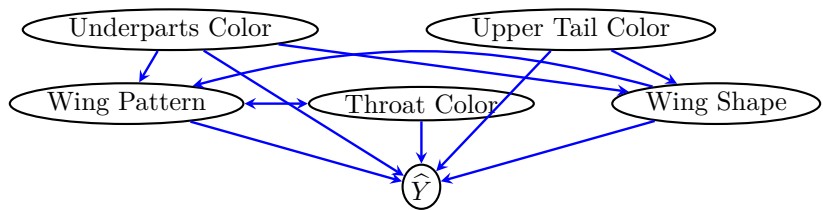

Figure 9: A PAG (subgraph) estimated from the bird image data using FCI. $\widehat{Y}$ denotes the output of the bird classification neural network ($\widehat{Y}$ takes one of 9 possible outcome labels). For considerations of space and readability, we only display the subgraph over variables adjacent to $\widehat{Y}$ on this run of the algorithm. Directed and edges into $\widehat{Y}$ from several variables indicate that these are likely causes of the prediction algorithm's output, whereas many other variables are mostly not causally relevant. Across subsamples of the data, the graph varies and some edges are more stable than others: `Wing Shape` and `Wing Pattern` are picked out as the most stable causes of the algorithm's output.

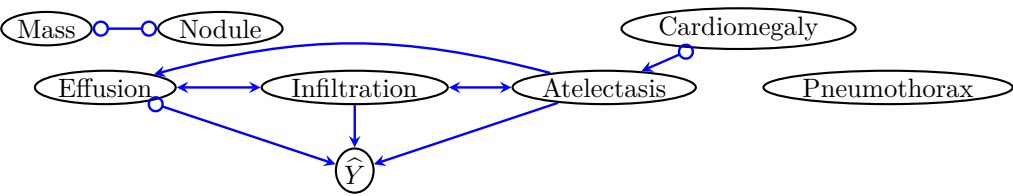

Figure 10: A PAG estimated from the X-ray data using FCI. $\widehat{Y}$ denotes the output of the pneumonia classification neural network ($\widehat{Y} = 1$ for pneumonia cases, $\widehat{Y} = 0$ for controls). Directed and partially directed edges into $\widehat{Y}$ from `Atelectasis`, `Infiltration`, and `Effusion` indicate that these are likely causes of the prediction algorithm's output, whereas the other variables are mostly not causally relevant. Across subsamples of the data, edges from these three key variables are sometimes directed and sometimes partially directed.

To compare our proposal with LIME and SHAP, we apply LIME and SHAP to the same neural network training pipeline used in our bird classification and X-ray experiments. We use implementations of LIME and SHAP available online.[2] For SHAP we apply the Deep Explainer module and for both algorithms we use default settings in the software. Since LIME and SHAP provide local explanations (in contrast with the type-level causal explanations provided by our approach), we present some representative examples from output on both classification tasks and display these in Figures 11 and 12. LIME and SHAP seem to highlight "important" pixels corresponding to the bird or the X-ray, but they do not communicate or distinguish what is important about the highlighted regions. For example, if the body of a bird is (in part or entirely) highlighted, a user cannot distinguish on the basis of LIME whether it is the wing pattern, wing shape, size, or some other relevant ornithological feature that is underlying the algorithm's classification. All one can conclude is that the "bird-body-region" of the image is important for classification, which is a reasonable baseline requirement for any bird classification algorithm but provides limited insight. Similarly, SHAP highlights pixels or clusters of pixels, but the pattern underlying these pixels and their relationship to bird characteristics are not transparent or particularly informative. In the X-ray images, we see the same issues: the diffuse nature of potential lung disease symptoms (varieties of tissue damage, changes in lung density, presence of fluid) is not clearly distinguished by either LIME or SHAP. Both algorithms simply highlight substantial sections of the lung. There is a mismatch between what is scientifically interpretable in these tasks and the "interpretable features" selected by LIME and SHAP, which are super-pixels or regions of pixel space. Finally, the local image-specific output of these algorithms makes it difficult to generalize about what in fact is "driving" the behavior of the neural network across the population of images. We conclude that LIME and SHAP, though they may provide valuable information in other settings, are largely uninformative relative to our goals in these image classification tasks in which distinguishing between spatially co-located

---

[2]Available at `https://github.com/marcotcr/lime` and `https://github.com/slundberg/shap`.

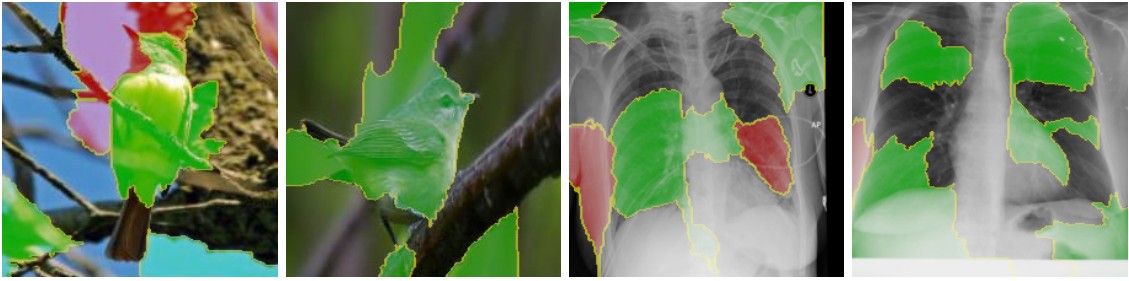

Figure 11: Results for LIME on selected images from the birds and X-ray datasets. Green highlights super-pixels with positive contribution to the predicted class and red indicates negative contribution.

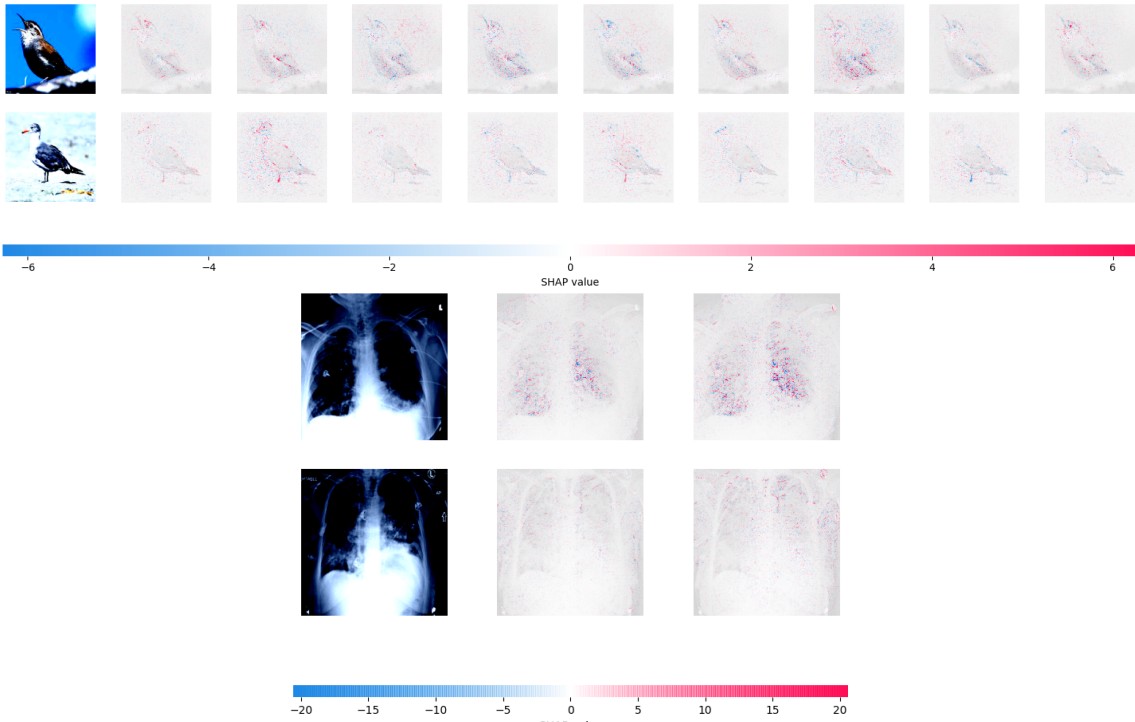

Figure 12: Results for SHAP on selected images from the birds and X-ray datasets. Columns correspond to output class labels (i.e., 9 bird categories and the binary pneumonia label). Original input images are in the leftmost column. Large SHAP values indicate strong positive (red) or negative (blue) contributions to the predicted class label.

image characteristics ("wing shape" and "wing pattern" are both co-located on the wing-region of the image, pneumothorax and effusion are distinct problems that may affect the same region of the chest) is important to support meaningful explanatory conclusions.

In contrast, the graphical representations learned by FCI do distinguish between meaningfully distinct causal factors that may occupy roughly the same region of pixel space. Namely, in the bird classification task our results suggest that `Wing Shape` and `Wing Pattern` are important determinants of the output but `Wing Color` is not, `Tail Shape` may be important but not `Tail Pattern`, etc. In the X-ray experiment, FCI identifies clinical symptoms such as `Infiltration`, `Atelectasis`, and `Effusion` as important, but not the presence of a `Mass` or lung `Nodule`. Without such interpretable features, a highlighted region of lung tissue may leave it ambiguous which kind of abnormality affecting that region is causally relevant to the classifier. The ability to distinguish between "similar" and spatially-overlapping causal factors is a consequence of

using clinically distinguishable interpretable features as input to the structure learning algorithm – for this reason, we emphasize that input and validation from domain experts is important for applying the proposed approach in practice (and we return to this issue below). Whether algorithm end-users will find the distinctions relevant will depend on the domain and the degree of collaboration.

# 7    Discussion

We have presented an analytical approach (based on existing tools) to support explaining the behavior of black-box prediction algorithms. Below we discuss some potential uses and limitations.

## 7.1    Algorithm Auditing and Evaluation

One important goal related to building explainable AI systems is the auditing and evaluation of algorithms post-hoc. If a prediction algorithm appears to perform well, it is important to understand why it performs well before deploying the algorithm. Users will want to know that the algorithm is "paying attention to" the right aspects of the input and not tracking spurious artifacts (Bhatt et al., 2020). This is important both from the perspective of generalization to new domains as well as from the perspective of fairness. To illustrate the former, consider instances of "dataset bias" or "data leakage" wherein an irrelevant artifact of the data collection process proves very important to the performance of the prediction method. This may impede generalization to other datasets where the artifact is absent or somehow the data collection process is different. For example, Winkler et al. (2019) study the role of violet image highlighting on dermoscopic images in a skin cancer detection task. They find that this image highlighting significantly affects the likelihood that a skin lesion is classified as cancerous by a commercial CNN tool. (The researchers are able to diagnose the problem because they have access to the same images pre- and post-highlighting: effectively, they are able to simulate an intervention on the highlighting.)

To illustrate the fairness perspective, consider a recent episode of alleged racial bias in Google's Vision Cloud API, a tool which automatically labels images into various categories (Kayser-Bril, 2020). Users found an image of a dark-skinned hand holding a non-contact digital thermometer that was labeled "gun" by the API, while a similar image with a light-skinned individual was labeled "electronic device." More tellingly, when the image with dark skin was crudely altered to contain light beige-colored skin (an intervention on "skin color"), the same object was labeled "monocular." This simple experiment was suggestive of the idea that skin color was inappropriately a cause of the object label and tracking biased or stereotyped associations. Google apologized and revised their algorithm, though denied any "evidence of systemic bias related to skin tone."

Auditing algorithms to determine whether inappropriate features have a causal impact on the output can be an important part of the bias-checking pipeline. Moreover, a benefit of our proposed approach is that the black-box model may be audited without access to the model itself, only the predicted values. This may be desirable in some settings where the model itself is proprietary.

## 7.2    Informativeness and Background Knowledge

It is important to emphasize that a PAG-based causal discovery analysis is informative to a degree that depends on the data (the patterns of association) and the strength of imposed background knowledge. Here we only imposed the minimal knowledge that $\widehat{Y}$ is not a cause of any of the image features and we allowed for arbitrary causal relationships and latent structure otherwise. Being entirely agnostic about the possibility of unmeasured confounding may lead, depending on the observed patterns of dependence and independence in the data, to only weakly informative results if the patterns of association cannot rule out possible confounding anywhere. If the data fails to rule out confounders and fails to identify definite causes of $\widehat{Y}$, this does not indicate that the analysis has failed but just that only negative conclusions are supported – e.g., the chosen set of interpretable features and background assumptions are not sufficiently rich to identify the causes of the output. It is standard in the causal discovery literature to acknowledge that the strength of supported causal conclusions depends on the strength of input causal background assumptions (Spirtes et al., 2000). In some cases, domain knowledge may support restricting the set of possible causal structures, e.g., when it

is believed the some relationships must be unconfounded or some correlations among $Z$ may only be due to latent variables (because some elements of $Z$ cannot cause each other).

### 7.3 Selecting or Constructing Interpretable Features

In our experiments we use hand-crafted interpretable features that are available in the form of annotations with the data. Annotations are not always available in applications. In such settings, one approach would be to manually annotate the raw data with expert evaluations, e.g., when clinical experts annotate medical images with labels of important features (e.g. "tumor in the upper lobe," "effusion in the lung," etc). In settings where annotations are available only on a small subset of examples, semi-supervised learning techniques developed in recent years may be useful for generating "pseudo-labels," and this may make the approach described here feasible even in settings with only a few annotated examples. Pipelines such as FixMatch (Sohn et al., 2020) use the predictions from a supervised model trained on annotated data combined with multiple "augmented" versions of unlabeled images that are expected to having matching labels to generate pseudo-labels at scale with excellent predictive performance.

Alternatively, one may endeavor to extract interpretable features automatically from the raw unlabeled data. Unsupervised learning techniques may be used in some contexts to construct features, though in general there is no guarantee that these will be substantively (e.g., clinically) meaningful or correspond to manipulable elements of the domain. We explored applying some recent proposals for automatic feature extraction methods to our problem, but unfortunately, none of the available techniques proved appropriate, for reasons we discuss.

A series of related papers (Chalupka et al., 2015; 2016; 2017) has introduced techniques for extracting *causal features* from "low-level" data, i.e., features that have a causal effect on the target outcome. In Chalupka et al. (2015), the authors introduce an important distinction underlying these methods: observational classes versus causal classes of images, each being defined as an equivalence class over conditional or interventional distributions respectively. Specifically, let $Y$ be a binary random variable and denote two images by $X = x, X = x'$. Then $x, x'$ are in the same *observational* class if $p(Y \mid X = x) = p(Y \mid X = x')$. $x, x'$ are in the same *causal* class if $p(Y \mid \mathrm{do}(X = x)) = p(Y \mid \mathrm{do}(X = x'))$. Under certain assumptions, the causal class is always a coarsening of the observational class and so-called "manipulator functions" can be learned that minimally alter an image to change the image's causal class. The idea is that relevant features in the image are first "learned" and then manipulated to map the image between causal classes. However, there are two important roadblocks to applying this methodology in our setting.

First, since our target behavior is the prediction $\widehat{Y}$ (which is some deterministic function of the image pixels), there is no observed or unobserved confounding between $\widehat{Y}$ and $X$. This implies our observational and causal classes are identical. Hence any manipulator function we learn would simply amount to making the minimum number of pixel edits to the image in order to alter its observational class, similar to so-called "minimum-edit" counterfactual methods (Wachter et al., 2017; Sharma et al., 2019; Goyal et al., 2019).

Second, even if we learn this manipulator function, the output is not readily useful for our purposes here. The function takes an image as input and produces as output another ("close") edited image with the desired observational class. It does not correspond to any label which we may apply to other images. (For example, taking a photo of an Oriole as input, the output would be a modified bird image with some different pixels, but these differences will not generally map on to anything we can readily identify with bird physiology and use to construct an annotated dataset.) This does not actually give us access to the "interpretable features" that were manipulated to achieve the desired observational class.

Alternative approaches to automatic feature selection also proved problematic for our purposes. In Lopez-Paz et al. (2017), the authors train an object detection network that serves as a feature extractor for future images, and then they use these extracted features to infer directions of causal effects. However, this approach relies on the availability of training data with a priori associated object categories: the Pascal Visual Object Classes, which are labels for types of objects (e.g., aeroplane, bicycle, bird, boat, etc.) (Everingham et al., 2015). This approach is thus not truly data-driven and relies on auxiliary information that is not generally available. In another line of work, various unsupervised learning methods (based on autoencoders or

variants of independent component analysis) are applied to extract features from medical images to improve classification performance (Arevalo et al., 2015; Sari & Gunduz-Demir, 2018; Jiang et al., 2018). These approaches are more data-driven but do not produce interpretable features – they produce a compact lower-dimensional representation of the pixels in an abstract vector space, which does not necessarily map onto something a user may understand or recognize. Recently, there has been interest in using large language models (LLMs) as annotators for unlabeled data, either alone or in collaboration with humans (Wang et al., 2024; Tang et al., 2024; Beck et al., 2025). This technology and its use for annotation is still in its development, but there is growing evidence that with careful engineering and human input, LLM-augmented annotations may cut costs for generating annotations at scale but also face limitations that are important to better understand.

In general, we observe a tradeoff between the interpretability of the feature set versus the extent to which the features are learned in a data-driven manner. Specifically, "high-level" features associated with datasets (whether they are extracted by human judgment or automatically) may not always be interpretable in the sense intended here: they may not correspond to manipulable elements of the domain. Moreover, some features may be deterministically related (which would pose a problem for most structure learning algorithms) and so some feature pre-selection may be necessary. Thus, human judgment may be indispensable at the feature-selection stage of the process and indeed this may be desirable if the goal is to audit or evaluate potential biases in algorithms as discussed in Section 7.1.

Finally, we note that macro-level feature annotations, whether they are derived from human judgment or statistical models, may contain errors. When labels are "noisy" or are corrupted by measurement error, this can degrade the performance of causal discovery algorithms. Some approaches have been proposed for mitigating the effect of measurement error on constraint-based algorithms (Blom et al., 2018), but in general this is a challenging and mostly open problem.

## 8 Conclusion

Causal structure learning algorithms – specifically PAG learning algorithms such as FCI and its variants – may be valuable tools for explaining black-box prediction methods. We have demonstrated the utility of using FCI in both simulated and real data experiments, where we are able to distinguish between possible causes of the prediction outcome and features that are associated due to unmeasured confounding. We hope the analysis presented here stimulates further cross-pollination between research communities focusing on causal discovery and explainable AI.

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

## A    Appendix: Calculations For Example Using Shapley Values

For the graph in Example 1, with the following SCM,

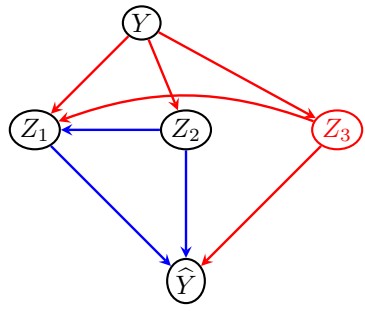

Figure 13: Data generating process for Example 1.

$$Y = \epsilon_Y$$
$$Z_3 = \gamma_0 + \gamma_1 Y + \epsilon_{Z_3}$$
$$Z_2 = \theta_0 + \theta_1 Y + \epsilon_{Z_2}$$
$$Z_1 = \nu_0 + \nu_1 Z_2 + \nu_2 Z_3 + \nu_3 Y + \epsilon_{Z_1}$$
$$\widehat{Y} = \mu_0 Z_1 Z_3 + \mu_1 Z_1 + \mu_2 Z_2$$

Example 1 assumes $Z_3$ is omitted, and backdoor adjustment is done using $Z_2$.

### A.1 Shapley Value With Valid Adjustment Set

The Shapley value for $Z_1$ is given as

$$\phi_{Z_1} = \frac{0!2!}{3!}\left\{f'_{\{Z_1\}} - f'_\phi\right\} + \frac{1!1!}{3!}\left\{f'_{Z_1,Z_2} - f'_{Z_2}\right\} + \frac{1!1!}{3!}\left\{f'_{Z_1,Z_3} - f'_{Z_3}\right\}$$
$$+ \frac{2!0!}{3!}\left\{f'_{\{Z_1,Z_2,Z_3\}} - f'_{\{Z_2,Z_3\}}\right\}$$

Computing each of the individual pieces

$$f'_\phi = \mathbb{E}[f'(Z_1, Z_2, Z_3)] = \mu_0\mathbb{E}[Z_1 Z_3] + \mu_1\mathbb{E}[Z_1] + \mu_2\mathbb{E}[Z_2]$$
$$f'_{\{Z_1\}} = \mathbb{E}[f'(z_1, Z_2, Z_3)] = \mu_0 z_1\mathbb{E}[Z_3] + \mu_1 z_1 + \mu_2\mathbb{E}[Z_2]$$
$$f'_{\{Z_1,Z_2\}} = \mathbb{E}[f'(z_1, z_2, Z_3)] = \mu_0 z_1\mathbb{E}[Z_3] + \mu_1 z_1 + \mu_2 z_2$$
$$f'_{\{Z_2\}} = \mathbb{E}[f'(Z_1, z_2, Z_3)] = \mu_0\mathbb{E}[Z_1 Z_3] + \mu_1\mathbb{E}[Z_1] + \mu_2 z_2$$
$$f'_{\{Z_1,Z_3\}} = \mathbb{E}[f'(z_1, Z_2, z_3)] = \mu_0 z_1 z_3 + \mu_1 z_1 + \mu_2\mathbb{E}[Z_2]$$
$$f'_{\{Z_3\}} = \mathbb{E}[f'(Z_1, Z_2, z_3)] = \mu_0 z_3\mathbb{E}[Z_1] + \mu_1\mathbb{E}[Z_1] + \mu_2\mathbb{E}[Z_2]$$
$$f'_{\{Z_1,Z_2,Z_3\}} = \mathbb{E}[f'(z_1, z_2, z_3)] = \mu_0 z_1 z_3 + \mu_1 z_1 + \mu_2 z_2$$
$$f'_{\{Z_2,Z_3\}} = \mathbb{E}[f'(Z_1, z_2, z_3)] = \mu_0 z_3\mathbb{E}[Z_1] + \mu_1\mathbb{E}[Z_1] + \mu_2 z_2$$

And this gives us the following Shapley value for $Z_1$

$$\phi_{Z_1} = \frac{1}{2}\left\{\mu_0 z_1\mathbb{E}[Z_3] - \mu_0\mathbb{E}[Z_1 Z_3] + \mu_0 z_1 z_3 - \mu_0 z_3\mathbb{E}[Z_1]\right\} + \mu_1\left\{z_1 - \mathbb{E}[Z_1]\right\}$$

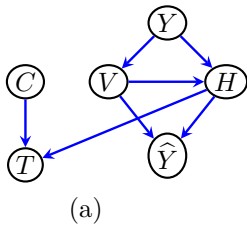

(a)

Figure 14: (a) A causal diagram representing the true data generating process utilized in the simulation study.

## A.2 Shapley Value With Invalid Adjustment Set

In this case, the Shapley value for $Z_1$ is

$$\phi_{Z_1} = \frac{0!1!}{2!}\left\{f'_{\{Z_1\}} - f'_\phi\right\} + \frac{1!0!}{2!}\left\{f'_{\{Z_1,Z_2\}} - f'_{\{Z_2\}}\right\}$$

Computing these with $Z_3$ missing

$$f'_\phi = \int f'(Z_1, Z_2, Z_3)p(Z_1, Z_2) = \mu_0 Z_3 \mathbb{E}[Z_1] + \mu_1 \mathbb{E}[Z_1] + \mu_2 \mathbb{E}[Z_2]$$

$$f'_{\{Z_1\}} = \int f'(z_1, Z_2, Z_3)p(Z_2) = \mu_0 z_1 Z_3 + \mu_1 z_1 + \mu_2 \mathbb{E}[Z_2]$$

$$f'_{\{Z_1,Z_2\}} = f'(z_1, z_2, Z_3) = \mu_0 z_1 Z_3 + \mu_1 z_1 + \mu_2 z_2$$

$$f'_{\{Z_2\}} = \int f'(Z_1, z_2, Z_3)p(Z_1) = \mu_0 Z_3 \mathbb{E}[Z_1] + \mu_1 \mathbb{E}[Z_1] + \mu_2 z_2$$

And so the incorrect Shapley value will be

$$\phi_{Z_1} = (z_1 - \mathbb{E}[Z_1])[\mu_0 z_3 + \mu_1]$$

## B  Appendix: Simulation Details And Implementation

HydraNet preloaded with ResNet-18 weights is fine tuned on a training dataset of 2500 images, with each image containing circles, triangles, horizontal bars and vertical bars all generated from an unbiased coin flip. The final layer of the network is replaced with 4 heads, each used to predict the presence of a circle, triangle, horizontal bar or vertical bar in the image. Each of the heads is a fully connected layer of size $(512, 2)$. All the parameters of this network except for the heads are frozen, and this network is optimized using a joint loss function as

$$loss = L_C + L_H + 2.5 * L_T + L_V$$

Where $L_C$ is the cross-entropy loss for the circle head, $L_H$ is the cross-entropy loss for the horizontal bar head, $L_T$ is the cross-entropy loss for the triangle head, and $L_V$ is the cross-entropy loss for the vertical bar head. The cross-entropy loss for the triangle head is up-weighted to encourage good triangle detection accuracy with HydraNet - earlier training runs where the weight was 1 did not have acceptable accuracy. The model is trained for 15 epochs with a batch size of 64 and using the SGD optimizer with a learning rate of 0.01 and a momentum of 0.09. Additionally, we schedule a learning rate decay with a step size of 7 and $\gamma = 0.1$.

We measure model performance on a validation dataset of 500 images, and our final network achieves the following accuracy on the validation dataset. A 100% accuracy in predicting circles, 99.60% for predicting

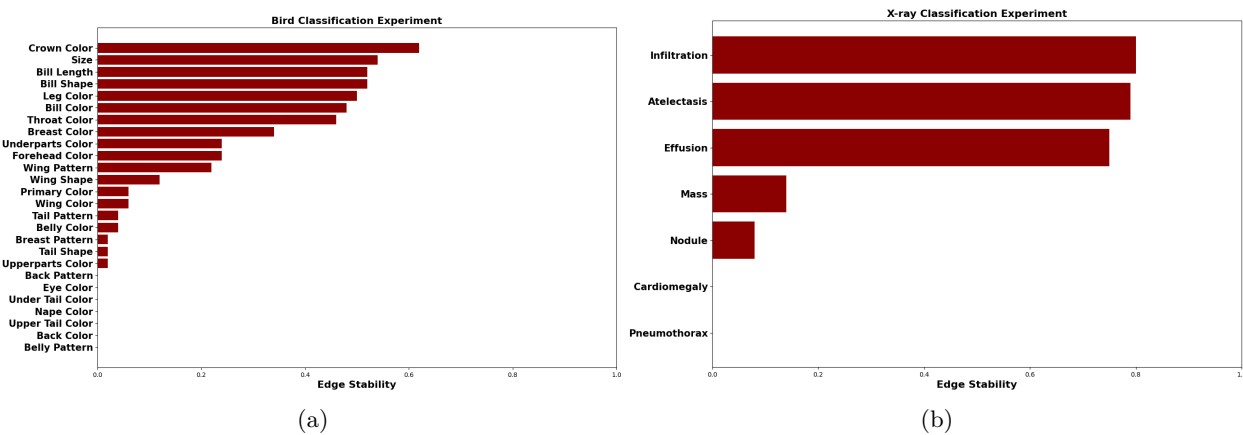

Figure 15: Results for two experiments using GRaSP-FCI. Potential causal determinants of (a) bird classifier output from bird images and (b) pneumonia classifier output from X-ray images.

horizontal bars, 97.80% accuracy for predicting triangles and 100% accuracy in predicting vertical bars. Our logistic regression model is using the sklearn library, with no penalty. The learned intercept is $-1.82530524$, and the learned coefficients are 0.09042825, 1.56043942, 0.07410451, and 1.27937188 for circle, horizontal bar, triangle and vertical bar respectively. This full black-box model achieves 77.89% accuracy on the dataset generated according to the DGP outlined in the simulation section.

For the FCI algorithm, we use the implementation in the TETRAD package, with the Chi-square independence test. Additionally, we impose the background knowledge that $\widehat{Y}$ must be a descendant of the interpretable features.

## C   Appendix: sensitivity analysis with hybrid causal discovery algorithm

As an alternative to FCI, we also apply a hybrid procedure called GRaSP-FCI to the data on chest X-ray and bird images. There is no fully greedy score-based procedure for learning PAGs analogous to the GES (Greedy Equivalence Search) algorithm for learning CPDAGs (Chickering, 2002). However, a two-step hybrid procedure has been proposed: first learn a CPDAG by using a greedy score based procedure, and then ignore edge directions and post-process the resulting undirected graph with some additional independence tests and FCI-orientation rules to produce a PAG (Ogarrio et al., 2016). Details underlying this hybrid approach and a proof of its correctness are described in Ogarrio et al. (2016). Though the authors use GES as the greedy score-based algorithm in that paper, more recently it has become possible to substitute the GRaSP (Greedy Relaxations of Sparsest Permutation) algorithm for learning a CPDAG (Lam et al., 2022) in the first stage of this two-step procedure; the result is called GRaSP-FCI. This algorithm is implemented in the TETRAD package and thus straightforward to run on the same data with the same pipeline (including the background knowledge described above). We use the BDeu score and Chi-square test of independence, with all other settings the same as for FCI. Results are summarized in Figure 15. In the X-ray image analysis, the ordering of features by edge stability is the same for GRaSP-FCI as FCI. The ordering for the bird image analysis has substantive differences: the edge stabilities are overall lower, and the top variables in the ordering are `Crown Color` and `Size`.

## D   Appendix: Preprocessing for Bird Dataset

For reproducibility, we describe our preprocessing of the bird image data.

### D.1  Bird Classification Grouping

The Caltech-UCSD 200-2011 Birds dataset comes with 200 categories of birds. We coarsen them using the group scheme below:

- Flycatcher
  - 041.Scissor_Tailed_Flycatcher
  - 040.Olive_Sided_Flycatcher
  - 043.Yellow_Bellied_Flycatcher
  - 038.Great_Crested_Flycatcher
  - 042.Vermilion_Flycatcher
  - 037.Acadian_Flycatcher
  - 039.Least_Flycatcher

- Gull
  - 059.California_Gull
  - 065.Slaty_Backed_Gull
  - 063.Ivory_Gull
  - 060.Glaucous_Winged_Gull
  - 066.Western_Gull
  - 062.Herring_Gull
  - 064.Ring_Billed_Gull
  - 061.Heermann_Gull

- Kingfisher
  - 081.Pied_Kingfisher
  - 082.Ringed_Kingfisher
  - 080.Green_Kingfisher
  - 079.Belted_Kingfisher
  - 083.White_Breasted_Kingfisher

- Sparrow
  - 127.Savannah_Sparrow
  - 126.Nelson_Sharp_Tailed_Sparrow
  - 116.Chipping_Sparrow
  - 114.Black_Throated_Sparrow
  - 121.Grasshopper_Sparrow
  - 119.Field_Sparrow
  - 122.Harris_Sparrow
  - 130.Tree_Sparrow
  - 128.Seaside_Sparrow
  - 118.House_Sparrow
  - 133.White_Throated_Sparrow
  - 115.Brewer_Sparrow
  - 117.Clay_Colored_Sparrow
  - 131.Vesper_Sparrow
  - 123.Henslow_Sparrow
  - 120.Fox_Sparrow

- – 129.Song_Sparrow
- – 125.Lincoln_Sparrow
- – 132.White_Crowned_Sparrow
- – 124.Le_Conte_Sparrow
- – 113.Baird_Sparrow

- Tern
  - – 141.Artic_Tern
  - – 144.Common_Tern
  - – 146.Forsters_Tern
  - – 143.Caspian_Tern
  - – 145.Elegant_Tern
  - – 147.Least_Tern
  - – 142.Black_Tern

- Vireo
  - – 156.White_Eyed_Vireo
  - – 154.Red_Eyed_Vireo
  - – 152.Blue_Headed_Vireo
  - – 157.Yellow_Throated_Vireo
  - – 153.Philadelphia_Vireo
  - – 155.Warbling_Vireo
  - – 151.Black_Capped_Vireo

- Warbler
  - – 168.Kentucky_Warbler
  - – 163.Cape_May_Warbler
  - – 158.Bay_Breasted_Warbler
  - – 162.Canada_Warbler
  - – 182.Yellow_Warbler
  - – 161.Blue_Winged_Warbler
  - – 167.Hooded_Warbler
  - – 180.Wilson_Warbler
  - – 171.Myrtle_Warbler
  - – 166.Golden_Winged_Warbler
  - – 173.Orange_Crowned_Warbler
  - – 177.Prothonotary_Warbler
  - – 178.Swainson_Warbler
  - – 169.Magnolia_Warbler
  - – 159.Black_and_White_Warbler
  - – 165.Chestnut_Sided_Warbler
  - – 164.Cerulean_Warbler
  - – 174.Palm_Warbler
  - – 176.Prairie_Warbler
  - – 160.Black_Throated_Blue_Warbler
  - – 179.Tennessee_Warbler
  - – 170.Mourning_Warbler
  - – 172.Nashville_Warbler

- – 181.Worm_Eating_Warbler
- – 175.Pine_Warbler

- Woodpecker

  - – 188.Pileated_Woodpecker
  - – 189.Red_Bellied_Woodpecker
  - – 190.Red_Cockaded_Woodpecker
  - – 187.American_Three_Toed_Woodpecker
  - – 191.Red_headed_Woodpecker,
  - – 192.Downy_Woodpecker

- Wren

  - – 195.Carolina_Wren
  - – 197.Marsh_Wren
  - – 196.House_Wren
  - – 193.Bewick_Wren
  - – 198.Rock_Wren
  - – 199.Winter_Wren
  - – 194.Cactus_Wren

Any other bird labels that did fit into these categories were excluded from our analysis.

### D.2 Attribute Groupings

The attributes were grouped as follows:

- bill_shape:

  - – 0 - Missing
  - – 1 - curved_(up_or_down), hooked, hooked_seabird
  - – 2 - dagger, needle, cone
  - – 3 - specialized, all-purpose
  - – 4 - spatulate

- wing_color:

  - – 0 - Missing
  - – 1 - blue, yellow, red
  - – 2 - green, purple, orange, pink, buff, iridescent
  - – 3 - rufous, grey, black, brown
  - – 4 - white

- upperparts_color:

  - – 0 - Missing
  - – 1 - blue, yellow, red
  - – 2 - green, purple, orange, pink, buff, iridescent
  - – 3 - rufous, grey, black, brown
  - – 4 - white

- underparts_color

  - – 0 - Missing

- 1 - blue, yellow, red
- 2 - green, purple, orange, pink, buff, iridescent
- 3 - rufous, grey, black, brown
- 4 - white

- breast_pattern

  - 0 - Missing
  - 1 - solid
  - 2 - spotted
  - 3 - striped
  - 4 - multi-colored

- back_color

  - 0 - Missing
  - 1 - blue, yellow, red
  - 2 - green, purple, orange, pink, buff, iridescent
  - 3 - rufous, grey, black, brown
  - 4 - white

- tail_shape

  - 0 - Missing
  - 1 - forked_tail
  - 2 - rounded_tail
  - 3 - notched_tail
  - 4 - fan-shaped_tail
  - 5 - pointed_tail
  - 6 - squared_tail

- upper_tail_color

  - 0 - Missing
  - 1 - blue, yellow, red
  - 2 - green, purple, orange, pink, buff, iridescent
  - 3 - rufous, grey, black, brown
  - 4 - white

- breast_color

  - 0 - Missing
  - 1 - blue, yellow, red
  - 2 - green, purple, orange, pink, buff, iridescent
  - 3 - rufous, grey, black, brown
  - 4 - white

- throat_color

  - 0 - Missing
  - 1 - blue, yellow, red
  - 2 - green, purple, orange, pink, buff, iridescent
  - 3 - rufous, grey, black, brown
  - 4 - white

- eye_color

    - 0 - Missing
    - 1 - blue, yellow, red
    - 2 - green, purple, orange, pink, buff, iridescent
    - 3 - rufous, grey, black, brown
    - 4 - white

- bill_length

    - 0 - Missing
    - 1 - about_the_same_as_head
    - 2 - longer_than_head
    - 3 - shorter_than_head

- forehead_color

    - 0 - Missing
    - 1 - blue, yellow, red
    - 2 - green, purple, orange, pink, buff, iridescent
    - 3 - rufous, grey, black, brown
    - 4 - white

- under_tail_color

    - 0 - Missing
    - 1 - blue, yellow, red
    - 2 - green, purple, orange, pink, buff, iridescent
    - 3 - rufous, grey, black, brown
    - 4 - white

- nape_color

    - 0 - Missing
    - 1 - blue, yellow, red
    - 2 - green, purple, orange, pink, buff, iridescent
    - 3 - rufous, grey, black, brown
    - 4 - white

- belly_color

    - 0 - Missing
    - 1 - blue, yellow, red
    - 2 - green, purple, orange, pink, buff, iridescent
    - 3 - rufous, grey, black, brown
    - 4 - white

- wing_shape

    - 0 - Missing
    - 1 - rounded-wings
    - 2 - pointed-wings
    - 3 - broad-wings
    - 4 - tapered-wings
    - 5 - long-wings

- size

  - 0 - Missing
  - 1 - large__(16-32in)
  - 2 - small__(5-9in)
  - 3 - very_large__(32-72in)
  - 4 - medium__(9-16in)
  - 5 - very_small__(3-5in)

- back_pattern

  - 0 - Missing
  - 1 - solid
  - 2 - spotted
  - 3 - striped
  - 4 - multi-colored

- tail_pattern

  - 0 - Missing
  - 1 - solid
  - 2 - spotted
  - 3 - striped
  - 4 - multi-colored

- belly_pattern

  - 0 - Missing
  - 1 - solid
  - 2 - spotted
  - 3 - striped
  - 4 - multi-colored

- primary_color

  - 0 - Missing
  - 1 - blue, yellow, red
  - 2 - green, purple, orange, pink, buff, iridescent
  - 3 - rufous, grey, black, brown
  - 4 - white

- leg_color

  - 0 - Missing
  - 1 - blue, yellow, red
  - 2 - green, purple, orange, pink, buff, iridescent
  - 3 - rufous, grey, black, brown
  - 4 - white

- bill_color

  - 0 - Missing
  - 1 - blue, yellow, red
  - 2 - green, purple, orange, pink, buff, iridescent
  - 3 - rufous, grey, black, brown

- – 4 - white

- crown_color
  - – 0 - Missing
  - – 1 - blue, yellow, red
  - – 2 - green, purple, orange, pink, buff, iridescent
  - – 3 - rufous, grey, black, brown
  - – 4 - white

- wing_pattern
  - – 0 - Missing
  - – 1 - solid
  - – 2 - spotted
  - – 3 - striped
  - – 4 - multi-colored

