# OpenReview forum: "Explaining the Behavior of Black-Box Prediction Algorithms with Causal Learning"
_TMLR — Accepted by TMLR_

### Review · Reviewer_mZit · 2025-02-14

**Summary Of Contributions:**

This paper focuses on the explainability of black-box machine learning models.
The authors propose a novel algorithm based on learning a causal graph between the predicted label and annotated high-level features.
The causal graph is learned with standard causal discovery methods like FCI.
Applied on two real-world datasets, a bird classification task and a medical diagnosis task, the authors show that the approach can identify the most relevant features for the prediction.

**Audience:**

Yes

**Broader Impact Concerns:**

No broader impact concerns were identified.

**Claims And Evidence:**

Yes

**Requested Changes:**

**Critical Changes**

I encourage the authors to include a more detailed discussion on the sample size needed for the approach, and its sensitivity to noisy expert annotations.
Further, a discussion on how to scale this approach to larger datasets, where manual annotations are not feasible, would be valuable.

**Minor Changes**

The highlighting in Figure 11 is done with a very low alpha, such that the actual content behind it is almost invisible. It would be better to use more transparent overlaps, such that the content is still visible.

The text in Figure 8 is very small. It would be better to increase the font size, such that the text is more readable.

**Strengths And Weaknesses:**

### Strengths

The paper presents an approach to explain the behavior of black-box prediction algorithms by identifying the features that are most relevant for the prediction.
The approach is based on causal learning and uses expert annotations to identify the features.
The authors show that the approach can be used to identify the most relevant features on two real-world datasets, a bird classification task and a medical diagnosis task.
The identified features are indeed relevant for the prediction, which shows that the approach can be used to explain the behavior of black-box prediction algorithms.
Overall, the approach can be a valuable tool for certain usecases, where expert annotations are available and the goal is to understand the behavior of the model.

The paper is well-written and easy to follow.
The explanations and introduction to topics are quite thorough, such that also readers not familiar with causal learning can follow the paper.
The experimental setup is detailed and the results are presented in a clear and understandable way.

### Weaknesses

The approach has the clear disadvantage of needing manual annotations of the features.
This requires an expert annotator for the task, which especially in the medical domain can be quite expensive.
Further, novel complex features, that may not be known to the annotator, are not considered in the analysis.
This is in contrast to existing methods, that give pixel-level explanations and not require manual annotations.

Additionally, it is unclear how many samples are needed for really stable results.
For example, for the bird classification task, a dataset of 3500 images was used.
But already there, a considerable amount of pre-processing was done on the expert-feature annotations to reduce the space of possible features.
This might not be feasible for smaller datasets, where the number of samples is limited.

---

> ### Author Response · Authors · 2025-03-03
> **Responses to reviewer mZit**
>
> •	*I encourage the authors to include a more detailed discussion on the sample size needed for the approach, and its sensitivity to noisy expert annotations.*
>
> Response: there are no known sample complexity results for the FCI algorithm, nor most causal discovery algorithms in general -- the theoretical properties of FCI (and variants) are asymptotic results (Colombo et al. 2012). The only related work we are aware of is a sample complexity result for the PC algorithm (Wadhwa and Dong 2021 arXiv:2102.03274) when variables are all discrete with $\ell$ categories and the maximum necessary conditioning set size is known (call this $m$); in that case, the number of samples necessary to achieve a certain error bound is in the worst-case is proportional to ${p \choose 2} \sum_{k=0}^m {p-2 \choose k}\ell^{3k/8}$. That is, the complexity is exponential in the number of variables $p$. Since FCI and PC share the same core steps, a similar but harder result should apply for FCI. This result is not very practically useful for selecting a sample size (it depends on unknown constants) but just confirms that conditional independence testing and causal discovery are statistically hard problems. Simulation experiments throughout the causal discovery literature indicate that it is very difficult to exactly recover the correct graph with any finite sample size, but that graphs with good precision can in many cases be estimated from on the order of a few thousand samples. There is unfortunately, no good general-purpose answer to “how many samples are needed for causal discovery.” There is also very little work on the effect of variable measurement error (noisy expert annotations) on causal discovery. One relevant work is Blom et al. (2018) which provides an adjustment for conditional independence tests when variables are subject to random error, though this focuses on continuous rather than discrete features. In the revised manuscript, we add some text to discuss the finite sample behavior of FCI in Section 4. We discuss measurement error and refer to Blom et al. (2018) at the end of Section 7.3.
>
> •	*Further, a discussion on how to scale this approach to larger datasets, where manual annotations are not feasible, would be valuable.*
>
> Response: we add a discussion in Section 7.3 on how semi-supervised learning and LLM technology could assist in automatic annotation of large datasets. Specifically, we add the following text: “In settings where annotations are available only on a small subset of examples, semi-supervised learning techniques developed in recent years may be useful for generating “pseudo-labels,” and this may make the approach described here feasible even in settings with only a few annotated examples. Pipelines such as FixMatch (Sohn et al., 2020) use the predictions from a supervised model trained on annotated data combined with multiple “augmented” versions of unlabeled images that are expected to having matching labels to generate pseudo-labels at scale with excellent predictive performance.” AND “Recently, there has been interest in using large language models (LLMs) as annotators for unlabeled data, either alone or in collaboration with humans (Wang et al., 2024; Tang et al., 2024; Beck et al., 2025). This technology and its use for annotation is still in its development, but there is growing evidence that with careful engineering and human input, LLM-augmented annotations may cut costs for generating annotations at scale but also face limitations that are important to better understand.”
>
> *Minor changes to figures*
>
> Response: the suggested changes are implemented in the revision

---

### Review · Reviewer_c3wj · 2025-02-17

**Summary Of Contributions:**

In this paper, the authors propose a causal approach to post-hoc explainability for black-box models, addressing two key issues: the reliance on micro-level features (e.g., pixels) instead of interpretable macro-level features and the unrealistic assumption of no unmeasured confounding. The authors introduce a causal graphical method that differentiates between features that causally influence model predictions and those merely associated with confounding. The authors evaluate the proposed method on the simulations and real-world datasets, enabling more meaningful explanations for black-box models without requiring access to their internal workings.

**Audience:**

Yes

**Broader Impact Concerns:**

N.A.

**Claims And Evidence:**

No

**Requested Changes:**

N.A.

**Strengths And Weaknesses:**

However, there are some problems to solve.
1.	The motivation of this paper is unclear. For example, why is it necessary to distinguish between variables with direct causal effects and those influenced by confounders?
2.	How are micro-level and macro-level features defined?
3.	While post-hoc explainability is valuable, if the macro-level latent variables in the model are not distinguishable (e.g., features with semantic confusion), is the proposed method still applicable?
4.	One aspect is particularly confusing: z is essentially already a latent variable, yet the authors introduce unmeasured confounding on top of z. Does this imply that the latent variables have not been fully recovered? Furthermore, not all black-box models align with the true data generation process, meaning there is no guarantee that z in a black-box model corresponds to the latent variables in the real data process. As a result, the model's explanations could be misleading.
5.	More causal explainability work needs to be considered and discussed, such as [1][2][3].

[1] Chen, Xuexin, et al. "Feature Attribution with Necessity and Sufficiency via Dual-stage Perturbation Test for Causal Explanation." arXiv preprint arXiv:2402.08845 (2024).
[2] Beckers, Sander. "Causal explanations and XAI." Conference on causal learning and reasoning. PMLR, 2022.
[3] Venkateswaran, Aparajithan, and Emilija Perković. "Towards complete causal explanation with expert knowledge." arXiv preprint arXiv:2407.07338 (2024).

---

> ### Author Response · Authors · 2025-03-03
> **Responses to reviewer c3wj**
>
> *Why is it necessary to distinguish between variables with direct causal effects and those influenced by confounders?*
>
> Response: on the theory of causal explanation which motivates our approach, discussed in Section 2, to “explain” the output of a model requires knowing which variables could be manipulated by intervention to change the model output. We add the following text to the introduction to clarify this: “Distinguishing whether some variable is a cause of model outputs rather than only associated due to confounding is crucial to understanding whether manipulations of that feature will make any difference to the prediction model output; intervening on variables associated only due to confounding will have no consequence and provides no explanation of the outcome.”
>
> *How are micro-level and macro-level features defined?*
>
> Response: we add the following text to the manuscript introduction to clarify this: “We use “micro-level” to refer to the raw features directly input into the prediction model, which are typically pixel or voxel values in image analysis and may be words or tokens in text analysis. “Macro-level” features then correspond to (observed or latent) abstractions of the raw features that are independently manipulable, also described as causal representations or disentangled representations (Scholkopf et al. 2021).” In Section 4 the model assumptions are described formally using structural equations.
>
> *While post-hoc explainability is valuable, if the macro-level latent variables in the model are not distinguishable (e.g., features with semantic confusion), is the proposed method still applicable?*
>
> Response: If the reviewer has in mind here that both (1) semantically-meaningful macro-level variables are not known and (2) not able to be learned, then the proposed approach is indeed not applicable. In that case, it is not clear to us how a causal explanation of the model’s outputs using any method could proceed, since the relevant explanatory units seem out of reach. However, disentangled representation learning is an active field of research, and so we expect that in cases of interest, learning macro-level variables may be possible.
>
> *The authors introduce unmeasured confounding on top of z. Does this imply that the latent variables have not been fully recovered?*
>
> Response: thank you for allowing us to clarify this important point. No latent variables are being recovered. The setting assumed here (formally described in Section 4) is that some variables in Z are observed and some are latent, i.e., images are annotated with some macro-level features (wing shape, throat color of birds, radiological findings on the x-rays) but these may not correspond exactly to the true macro-level representations used in the black-box model, which are all some unknown functions of the micro-level features X. The goal here is not to recover or estimate any latent features, but to determine whether some user-specified features in Z are in fact causes of the model’s output, or not. If these variables are not causes but are statistically associated with model’s output, it is because there is confounding.
>
> *Additional references [1][2][3]*
>
> Response: the revised manuscript cites and discusses these additional works. The first reference [1] is now cited alongside other “perturbation-based” approaches to XAI in Section 3.3. The second reference [2] is mentioned in Section 2. To clarify, the last work [3] is not related to XAI or causal explainability in the sense relevant here – it presents formal results for reasoning with maximal ancestral graphs given background knowledge about edge orientations. (This is related to development of new causal discovery algorithms but not the present manuscript.)

---

### Review · Reviewer_GYZw · 2025-02-18

**Summary Of Contributions:**

This submission proposes to use the FCI algorithm to estimate a PAG over a set of macro-level variables that is correct even if there exist confounders that are not in the set of macro-level variables. It applies this technique to a classifier trained on synthetic data, where it is shown to correctly infer the PAG when one of the true causal factors is omitted from the set of explanatory features, and to bird and chest x-ray classifiers.

**Audience:**

Yes

**Claims And Evidence:**

No

**Requested Changes:**

- The submission needs experiments that show that FCI is capable of correctly inferring the PAG in nontrivial settings. The current experiments show that FCI can correctly infer the PAG in a somewhat trivial setting and that it infers a PAG in real settings, but it is unclear if the inferred PAG in these real settings is correct. To demonstrate correctness, the authors might consider perturbing some of the inferred causal determinants to verify the inferred PAG, e.g., showing that the bird classifier is sensitive to throat color but invariant to bill color. Alternatively, it might be possible to train classifiers on intentionally confounded real-world data, determine the features the classifier uses by testing it on unconfounded data, and then analyze the ability of causal discovery methods to discover those features.
- It would be nice to have a more detailed description of FCI and the situations under which it is and is not effective. In the current submission, it feels like it's a magical algorithm that can infer PAGs in the presence of unmeasured confounding, but given the challenges of causal inference, I suspect that it has limitations.
- It would be nice to see a comparison of FCI with PC and perhaps other causal discovery methods, to see if the conclusions are substantially different.
- (Non-essential) Although the submission is a pleasure to read, I feel that it could be shortened without compromising essential detail.

**Strengths And Weaknesses:**

Strengths:
- The motivation for the proposed method is clearly described and seems reasonable.
- The proposed method is robust to confounders even when the practitioner has not included these confounders in the set of candidate explanatory features.

Weaknesses:
- At 16 pages, the submission is quite long, with extensive discussion of the philosophy of causal inference and how the proposed method might be applied, but includes less than a paragraph of detail of the the FCI algorithm itself, which is core to the proposed method.
- There is no obvious ground truth for the causal features of the bird and pneumonia classifiers, so we do not know whether the inferred causal features used by the bird and pneumonia classifiers are actually the correct causal features. The finding that features associated with pneumonia seem to be causal determinants of pneumonia is promising, but not conclusive.
- The experiments do not provide a quantitative measure of the effectiveness of the proposed method versus others. The only comparison with other methods is a demonstration that, when applied to pixels, explainability methods LIME and SHAP fail to provide interpretable explanation of the bird and chest x-ray classifiers. In Section 3.3, the submission mentions that there are other causality-aware feature importance approaches that can be applied to macro-level features (though it states that they cannot account for unmeasured confounding) but none of these methods are compared with FCI.
- Part of the introduction is a verbatim restatement of the abstract, which seems unnecessary.

---

> ### Author Response · Authors · 2025-03-03
> **Responses to reviewer GYZw**
>
> •	*The submission needs experiments that show that FCI is capable of correctly inferring the PAG in nontrivial settings. The current experiments show that FCI can correctly infer the PAG in a somewhat trivial setting and that it infers a PAG in real settings, but it is unclear if the inferred PAG in these real settings is correct.*
>
> Response: here the reviewer is quite reasonably touching on a fundamental challenge in causal discovery: real-world validation. The fact is that in settings where causal discovery is needed, no causal ground truth is known, and so there is no simple way to establish that an inferred graph is correct. There has been much discussion in the literature on causal discovery on this issue, including proposals on how to validate methods with semi-synthetic (but not fully real-world) data. In practice, most causal discovery papers do a version of what we do here: show correctness in simulated data where the truth is known and informally assess the plausibility of inferred graphs from real data where the truth is not known. (The exception is in some genomic or neuroscientific studies where some partial biological knowledge can be brought to bear on the quality of the inferred graph. Very rarely follow-up experiments/interventions such as gene knockouts can be carried out, see for example Stekhoven et al. 2012. This requires specialized technology, careful experimental design, and significant resources in the biological domain.)
> In our real data applications here (birds and x-rays), we are not aware of a way to straightforwardly validate the inferred PAGs. The reviewer’s suggestion to modify images and re-classify them using the CNN is one we did consider. However, we would need a pipeline for generating “modified” images that respect certain important constraints (e.g., birds that are otherwise very similar to real birds in the training data, where natural causal relationships among features are reproduced, but with *only* different throat color or bill shape  – this is not as straightforward as simply feeding images of “natural” birds with different coloration). Although it may be possible to accomplish this with VAEs or recent generative image technology, it would take substantial engineering to implement this correctly. Unfortunately, we do not have sufficient expertise in synthetic image generation technology, and we view building this kind of validation pipeline outside the scope of the present project, which has the more limited aim of motivating and illustrating a novel application of causal discovery in XAI.
>
> •	*It would be nice to have a more detailed description of FCI and the situations under which it is and is not effective.*
>
> Response: we are happy to add more detailed discussion of FCI to address this point. FCI is an algorithm proposed more than 20 years ago (Spirtes et al. 2000) and FCI itself is not part of the contributions of the present work, so some of the discussion of its properties is left to cited literature. In Section 4 we add a long paragraph describing the steps of the FCI algorithm and discussing its known properties.
>
> •	*It would be nice to see a comparison of FCI with PC and perhaps other causal discovery methods*
>
> Response: we agree, comparing FCI with another causal discovery algorithm may be illuminating. Since the key characteristic of our proposal is to allow for unmeasured confounding and learn a PAG (not a DAG), we chose in the revision to compare with another PAG-learning algorithm called GRaSP-FCI. (PC would not be able to distinguish between causal features and features associated due to unmeasured confounding. Moreover, the first step of FCI is identical to PC so the adjacencies would be basically the same.) FCI is a constraint-based method, but GRaSP-FCI is hybrid score-based method that ultimately estimates a PAG, and so it is worth understanding if results may vary with this alternative choice of discovery procedure. In particular, GRaSP-FCI takes an approach that begins by searching over variable permutations to optimize a version of the BIC score. This is the GRaSP algorithm, which achieves state-of-the-art performance in a setting with no unmeasured confounding (Lam et al. 2023). Then in the second stage, GRaSP-FCI implements additional tests and PAG-orientation rules to transform the result into a PAG. We mention this additional analysis in the main text and describe it in more detail in the appendix. The results can be summarized as follows: 1) for the chest X-ray experiment, results from GRaSP-FCI are substantively identical to FCI: the final variable ordering by “edge stability” is unchanged. 2) for the birds experiment, the results from GRaSP-FCI have some substantive differences in the final “edge stability” ordering – a few variables marked as important possible causes by FCI are downgraded by GRaSP-FCI and vice versa. But the algs do agree about most of the 27 variables. See Sec 6.3 and the additional figs reported in the Appendix.

---

### Author Response · Authors · 2025-03-03
**Thank you to reviewers**

We thank all the reviewers for their attention to our paper and insightful constructive comments. We addressed all the comments in our responses and uploaded a revision of the manuscript with requested changes (in blue text).

---

### Decision · Action_Editor_4NkD · 2025-04-11

**Recommendation:** Accept as is

**Comment:**

The main criticism is on the empirical evaluation, which contains real data sets but lacks causal ground truth. I took a closer look, and the results are reasonable, exceeding the standard of the community. The synthetic data, albeit simple, validates the claim, and the real data shows applicability to realistic scenarios.

**Audience:**

The paper is likely to be interesting for two subcommunities: those interested in causality research and its applications, as well as those interested in explainability and accountability of black box models.

**Claims And Evidence:**

This paper provides causal explanations in prediction settings using black box models, allowing for hidden confounding and macro-level features. The method is based on FCI, and is a novel application with convincing empirical evidence. It also nicely motivates the problem in the context of the causal literature and causal thinking, which I believe will be valuable for future works.